# A New Perspective: Revealing the Algicidal Properties of *Bacillus subtilis* to *Alexandrium pacificum* from Bacterial Communities and Toxins

**DOI:** 10.3390/md20100624

**Published:** 2022-09-30

**Authors:** Ruihong Cheng, Xiuxian Song, Weijia Song, Zhiming Yu

**Affiliations:** 1Key Laboratory of Marine Ecology and Environmental Sciences, Institute of Oceanology, Chinese Academy of Sciences, Qingdao 266071, China; 2Laboratory of Marine Ecology and Environmental Science, Qingdao National Laboratory for Marine Science and Technology, Qingdao 266237, China; 3University of Chinese Academy of Sciences, Beijing 100049, China; 4Center for Ocean Mega-Science, Chinese Academy of Sciences, Qingdao 266071, China

**Keywords:** algicidal bacteria, algicidal effect, paralytic shellfish toxins, *Alexandrium pacificum*, *Bacillus*, harmful algal blooms

## Abstract

Algicidal bacteria are important in the control of toxic dinoflagellate blooms, but studies on the environmental behavior of related algal toxins are still lacking. In this study, *Bacillus subtilis* S3 (S3) showed the highest algicidal activity against *Alexandrium pacificum* (Group IV) out of six *Bacillus* strains. When treated with 0.5% (*v*/*v*) S3 bacterial culture and sterile supernatant, the algicidal rates were 69.74% and 70.22% at 12 h, respectively, and algicidal substances secreted by S3 were considered the mechanism of algicidal effect. During the algicidal process, the rapid proliferation of *Alteromonas* sp. in the phycosphere of *A. pacificum* may have accelerated the algal death. Moreover, the algicidal development of S3 released large amounts of intracellular paralytic shellfish toxins (PSTs) into the water, as the extracellular PSTs increased by 187.88% and 231.47% at 12 h, compared with the treatment of bacterial culture and sterile supernatant at 0 h, respectively. Although the total amount of PSTs increased slightly, the total toxicity of the algal sample decreased as GTX1/4 was transformed by S3 into GTX2/3 and GTX5. These results more comprehensively reveal the complex relationship between algicidal bacteria and microalgae, providing a potential source of biological control for harmful algal blooms and toxins.

## 1. Introduction

Harmful algal blooms (HABs) are a common ecological disaster throughout the world. In particular, toxic dinoflagellate blooms have caused severe damage to the structure and function of aquatic ecosystems and economic losses to the tourism and aquaculture industries, and seriously endangered human health [1,2]. The *Alexandrium tamarense* Balech species complex is a group of toxic dinoflagellates that are among the most widely distributed HAB-causing taxa globally [3]. In particular, *Alexandrium pacificum* (*A. tamarense* Group IV) is widely distributed on the Chilean, French, Chinese, Japanese, and Korean coasts [4,5,6,7,8] and produces paralytic shellfish toxins (PSTs). PSTs are known to selectively block sodium channels, resulting in their inability to form action potentials and inhibit nerve conduction [9,10]. Humans can quickly experience numbness, dizziness, nausea, and diarrhea after eating contaminated seafood [11]. Numerous fatal cases of PSTs have been reported globally [12]. Toxic blooms with PSTs in Basha in 2013 have resulted in the intoxication of 58 people with 4 fatalities (records from the Sabah Health Department, Malaysia) [13]. With continuous research on the algae–bacteria relationship, bacteria with algicidal activity have been shown to play an important role in controlling HABs by lysing algal cells [14,15]. Commonly reported algicidal bacteria include *Pseudomonas* sp., *Alteromonas* sp., *Vibrio* sp., *Bacillus* sp., and *Planomicrobium* sp [14]. Most of these algicidal bacteria are isolated from the water environments of algal blooms or bacteria in the phycosphere [16,17]. However, this method of obtaining algicidal bacteria has some disadvantages, such as limited sources, as approximately 95% of marine bacteria are considered to be unculturable [18]. More importantly, the ecological impact of these algicidal bacteria will need to be evaluated in the future.

Contrary to the natural screening of bacteria, commercial bacteria have emerged as a potential source of algicidal bacteria because of their various advantages, including ease of availability, mass cultivation, and environmental friendliness. It should be noted that *Bacillus* sp. is widely used as a probiotic for aquaculture and water quality regulation because of its particular characteristics, such as inhibition of pathogenic bacteria [19], degradation of organic matter [20], improvement in animal immunity [21], and ability to survive in extreme environments [22]. *Bacillus* sp. has also been shown to have high algicidal activity against some species of HAB organisms, including cyanobacteria, dinoflagellates, and diatoms [16,23,24]. Therefore, to quickly obtain green and efficient algicidal bacteria, this paper attempts to screen an algicidal bacterium against *A. pacificum* from different commercial *Bacillus* strains.

Although some studies have been conducted to control *Alexandrium* blooms with algicidal bacteria *Vibrio* DHQ25, *Pseudoalteromonas* DHY3, and *Shewanella* Y1 [25,26], there has been no report of changes in PSTs during the algicidal process. However, PSTs, which are secondary metabolites synthesized in algal cells, are released into the water environment when the cells are lysed, causing direct and indirect harm to marine organisms as well as humans [27,28,29]. Therefore, it is imperative to investigate the changes in the content and toxicity of PSTs during the algicidal process by algicidal bacteria against typical toxic dinoflagellates to better comprehend their algicidal properties.

The purpose of this paper is to screen out an effective algicidal bacterium against *A. pacificum* from various *Bacillus* sp. strains and investigate the mechanisms of its algicidal effects. The changes in the bacterial community in the phycosphere and PSTs during the algicidal process were also studied to reveal the algicidal properties of the algicidal bacteria, providing a scientific basis for the microbial treatment of *A. pacificum* blooms.

## 2. Results

### 2.1. Algicidal Bacteria Screening

To screen out an effective algicidal bacterium against *A. pacificum* from various *Bacillus* sp. Strains—*B. pumilus*, *B. laterosporus*, *B. cereus*, *B. amyloliquefaciens*, *B. subtilis* S2, and *B. subtilis* S3 were inoculated into LB liquid medium for 24 h to obtain six kinds of bacterial culture (the bacterial culture contains bacteria pellets, bacterial secretions, and a small amount of residual medium). As shown in Figure 1, the majority of the algae cells changed from vegetative cells to pellicle cysts or lysed after 12 h of treatment with the 1% (*v*/*v*) bacterial culture from the six strains of *Bacillus* sp. There was a difference in algicidal activity between the six strains of *Bacillus* sp. against *A. pacificum*. Neither *B. amyloliquefaciens* nor *B. laterosporus* exhibited any algicidal activity on the algal cells. For *B. pumilus*, *B. cereus*, *B. subtilis* S2, and *B. subtilis* S3, the rates of lysed cells were 52.00%, 33.33%, 63.16%, and 73.12%, respectively. Therefore, *B. subtilis* S3 (hereafter referred to as S3) was selected as the algicidal bacterium against *A. pacificum* for further investigation.

### 2.2. Algicidal Effect of B. subtilis S3 on A. pacificum

The S3 bacterial culture and its components, including the bacterial resuspension, sterile supernatant, and LB liquid medium, were examined for algicidal activity against *A. pacificum*. As shown in Figure 2a, the algicidal activity of the S3 bacterial culture was correlated with the treatment concentration and time. The algicidal rates at 3 h reached 31.25%, 45.16%, 49.12%, and 57.26% for the 0.3%, 0.5%, 1%, and 2% S3 bacterial culture treatment groups, respectively. At 12 h, the algicidal rate for the 2% treatment group reached 80.23%, and the algicidal rates of the 0.5% and 1% treatment groups were 69.74% and 70.97%, respectively. The algicidal rate of the 0.3% treatment group was 45.16%, which was significantly lower than that of the other concentration experimental groups (*p* < 0.05). Additionally, various concentrations of sterile supernatant showed similar algicidal activity to that of *A. pacificum* as the bacterial culture (Figure 2c). However, neither the bacterial resuspension nor the LB liquid medium showed algicidal activity against *A. pacificum* within 12 h (Figure 2b,d).

### 2.3. Observation of the A. pacificum Cell Morphology

To investigate the algicidal process and mechanism of S3 on *A. pacificum*, we observed the morphological changes in the algal cells during the algicidal process by optical microscopy and scanning electron microscopy. As shown in Figure 3, the structure of control cells (Figure 3A) was intact after immediate exposure to 0.5% S3 bacterial culture for 0 h; then, the vegetative cells tended to shed their theca to form pellicle cysts with increasing treatment time. Because of the different states of the vegetative cells, three morphological changes occurred during the shedding process. First, approximately 45% of vegetative cells were seriously destroyed in organelles and membranes (Figure 3B) during the shedding process. Second, cytoplasmic condensation occurred in approximately 25% of vegetative cells after shedding their theca (Figure 3C,D), followed by complete lysis into cell fragments at 12 h (Figure 3E). Third, 30% of pellicle cysts (Figure 3F) formed successfully after vegetative cells shed their theca.

### 2.4. Changes in the Bacterial Community during the Algicidal Process

To further clarify the algicidal properties of *Bacillus subtilis* S3 fermentation broth, we investigated the effects of 0.5% (*v*/*v*) bacterial culture (group A), bacterial resuspension (group B), sterile supernatant (group C), and LB liquid medium (group D) on biological community and PSTs during the experiment, while untreated algal cultures were used as a control (group CK). As seen in Table 1, the bacterial abundance and community diversity in *A. pacificum* were evaluated after 12 h of treatment. The bacterial abundance of the CK group in the algal cultures was 1.01 × 10^7^ cells/mL, while the bacterial abundance in each treatment group increased by 1.3 to 1.8 times that of the CK at 12 h (*p* < 0.01). In addition, the Shannon and Simpson indices of groups A and B were 3.43 ± 0.41 and 0.83 ± 0.03, and 3.31 ± 0.39 and 0.82 ± 0.03, respectively, which were significantly higher than those of the CK group (*p* < 0.01). The Simpson index of group D was also significantly higher than that of the CK group (*p* < 0.05). 

As shown in Figure 4, high-throughput sequencing analysis of 16S rDNA revealed that the dominant genera in the bacterial community in the phycosphere of *A. pacificum* were *Marivita* sp. of the phyla α-Proteobacteria and *Phaeodactylibacter* sp. of the phylum Bacteroidetes, with relative abundance values of 47.71% and 27.48%, respectively. After each treatment was added at 0 h, the compositions of the bacterial communities in groups C and D did not change significantly compared with that in the CK group. In contrast, groups A and B visibly changed because of the addition of S3 cells, and *Bacillus* sp. emerged as the major dominant genus in the bacterial community. As the treatment time was extended, the composition of the bacterial community in each group changed little at 3 h compared with at 0 h. However, the relative abundance of *Marivita* sp. increased in groups A and B at 12 h. Importantly, at 12 h, the relative abundance of *Alteromonas* sp. increased by 7.81%, 8.68%, and 25.04% in groups A, C, and D containing exogenous nutrients, respectively, which was significantly higher than those in the CK and B groups (1.32% and 1.41%, respectively). 

### 2.5. Changes in the Contents and Toxicity of PSTs during the Algicidal Process

#### 2.5.1. Extracellular and Intracellular PSTs Contents

As shown in Figure 5a, the concentration of extracellular PSTs increased with prolonged treatment time over 12 h in groups A and C, where the algicidal effects occurred. The extracellular PSTs contents were 25.41 and 44.88 nmol/L in group A at 3 h and 12 h, respectively, which were significantly higher, by 62.97% (*p* < 0.05) and 187.88% (*p* < 0.01), compared with at 0 h. The extracellular PST contents in group C also showed similar changes to those in group A, with a significant increase of 231.47% (*p* < 0.01) at 12 h compared with that at 0 h. In contrast, the intracellular PST contents in groups A and C decreased with prolonged treatment time and decreased significantly, by 61.09% and 71.23%, at 12 h compared with at 0 h, respectively (*p* < 0.01, Figure 5b). However, in groups B and D, without algicidal activity, the contents of the extracellular and intracellular PSTs did not change significantly throughout the experimental period, except that the intracellular PSTs in group D were significantly increased, by 18.15%, at 12 h compared with at 0 h (*p* < 0.05).

#### 2.5.2. Total PSTs Content, Toxicity, and Profile

As shown in Figure 6a, the total PSTs contents in groups A and C gradually increased during the algicidal process and reached 57.80 and 58.26 nmol/L at 12 h, which were significantly increased, by 18.43% and 11.40%, compared with those at 0 h, respectively (*p* < 0.05). However, the total toxicity of the algal sample was significantly reduced by 15.40% and 12.08% (*p* < 0.01, Figure 6b). After analyzing the changes in each PSTs derivative, we found that GTX1/4 decreased from approximately 56% to 28% in groups A and C at 12 h, while GTX2/3 increased from approximately 4% to 20%, and GTX5 also increased by approximately 10% (Figure 6c and Appendix A). Although the proportions of each PST derivative in group D did not change significantly during the experiment, the total PST content and toxicity also increased significantly at 12 h. There were no clear changes in the toxicity, total content, or profile of PSTs in group B.

## 3. Discussion

### 3.1. Algicidal Effect of B. subtilis S3 on A. pacificum

In the continued study of algal–bacterial relationships, it was found that most algicidal bacteria exhibit relatively strict species specificity to microalgae [30,31]. In this paper, the different strains of *Bacillus* sp. also exhibited a certain species specificity toward *A. pacificum*. *B. amyloliquefaciens* and *B. lateralis* had no notable algicidal activity against *A. pacificum*, whereas *B. subtilis* S3 showed the highest algicidal activity. Under the experimental conditions of this study, the algicidal rate reached as high as 69.74% after treatment with 0.5% (*v*/*v*) S3 bacterial culture for 12 h, and the algicidal activity was positively correlated with the treatment concentration and time. The species specificity of *Bacillus* against *A. pacificum* may be attributed to the type and functions of the bacteria. Several studies have reported that the growth of *Microcystis aeruginosa* is inhibited by both *B. amyloliquefaciens* and *B. subtilis* [32,33], while only *B. subtilis* inhibits the growth of *Alexandrium minutum* [34]. Because of the intricate relationship between algae and bacteria, no studies have elucidated this species–specific mechanism.

In general, bacteria that inhibit the growth of algae work through direct [35] or indirect attacks. Indirect attacks, such as competition for nutrients or survival space with microalgae [36,37], secretion of algicidal substances [32,33], and flocculation of microalgae [38], account for approximately 70% [39] of growth-inhibiting actions. In this paper, we used the S3 bacterial culture, sterile supernatant, and bacterial resuspension to explore the mode of the algicidal effect of S3. The sterile supernatant was capable of lysing the algal cells as quickly as the bacterial culture, while the bacterial resuspension had no algicidal activity, suggesting that the main mode of the algicidal effect of S3 on *A. pacificum* was the secretion of algicidal substances. It was also found that *B. subtilis* can inhibit the growth of *M. aeruginosa* and *Cochlodinium polykrikoides* by secreting surfactin and peptides, respectively [33,40]. However, the species of algicidal substances secreted by S3 in this paper need further analysis and identification.

Algicidal substances are able to damage the structural integrity of algal cells by lysing their membranes [41] and destroying their chloroplasts [15] and nuclei [42]. As a result of our observations using optical microscopy and scanning electron microscopy, we found that the algicidal substances secreted by S3 also severely damaged the structure of *A. pacificum* cells. After the algal cells were treated with 0.5% S3 bacterial culture, clear morphology changes, such as cytoplasm shrinkage, organelle degradation, cell membrane disruption, and cell wall fragmentation, were observed. Furthermore, we observed that vegetative cells under the stress of S3 bacterial culture treatment tended to shed theca to form pellicle cysts, and the residual algal cells that did not lyse also presented as pellicle cysts. The pellicle cysts that form under natural conditions are considered by researchers to be part of the life cycle of algae, and these pellicle cysts could revert to vegetative cells when the conditions are suitable [43,44]. It is also possible that pellicle cysts may form under adverse conditions, such as low temperature [45], mechanical damage [45], chemical stress [45], nutrient deficiency [46], and algicidal bacteria exposure [47], to ensure their survival. Therefore, the main mechanism of the algicidal substance secreted by S3 may stress the vegetative cells to shed their theca and form pellicle cysts to protect themselves and, at the same time, exert the algicidal effect by destroying the structural integrity of algal cells.

### 3.2. Impact on Bacterial Community during the Algicidal Process

As algae grow, they release organic substances into their surroundings, which bacteria break down into inorganic substances to provide nutrients for the algae and their growth. In this unique exchange of material, energy, and information between algae and bacteria, the phycosphere is created [48]. As demonstrated in this study, *Marivita* sp. of the family Rhodobacteraceae and *Phaeodactylibacter* sp. of the family Saprospiraceae were the dominant genera in the bacterial community in the phycosphere of *A. pacificum*. Some bacteria affiliated with the marine family Rhodobacteraceae were found to produce auxins, vitamins, and siderophores, which are essential for establishing mutualistic relationships with algae [49]. Therefore, it could be speculated that *Marivita* sp. in the phycosphere may provide certain important growth factors to *A. pacificum.* There is a specific structure and function that bacteria and algae maintain in the phycosphere, which, when disturbed, either facilitates the growth of the microalgae or accelerates their death [37,50]. It was found that when a small amount of 2216E was added to nonaxenic and axenic *A. tamarense*, the growth of the axenic algae did not change, while all of the nonaxenic algae were lysed within 14 h. At this time, the abundance of *Alteromonas* sp. and *Thalassobius aestuarii* sp. increased significantly [50]. The bacterial abundance in *A. pacificum* also increased during the algicidal process with S3 bacterial culture and sterile supernatant. In particular, *Alteromonas* sp. preferentially utilize exogenous nutrients for its rapid proliferation, and its relative abundance increased by approximately 7% between 0 h and 12 h. Numerous studies have demonstrated that *Alteromonas* sp. has algicidal activity when its numbers exceed an algicidal threshold [14,17,51]. Consequently, it may be speculated that *Alteromonas* sp. that increased in our study may also have algicidal activity and could potentially contribute to the death of *A. pacificum* at a later stage during the algicidal process; however, further investigation is necessary to verify these conjectures. Although *Alteromonas* sp. quickly proliferated after treatment with LB liquid medium, the absence of algicidal activity was probably because the number of *Alteromonas* sp. did not reach the algicidal threshold.

### 3.3. Impact on PSTs during the Algicidal Process

#### 3.3.1. Extracellular and Intracellular PSTs

In the ocean, dinoflagellates, especially *Alexandrium*, are the main organisms capable of producing PSTs, and the mechanisms of PST synthesis and release are of great interest because of the major impact of PSTs on the ecosystem and human health [52,53]. PSTs are a class of water-soluble endogenous compounds that are released mainly upon algal death and lysis [54]. Therefore, in this study, we investigated the characteristics of PST release during the algicidal process based on *B. subtilis* S3 lysis of *A. pacificum*. Additionally, to the best of our knowledge, this is the first description of the changes in PSTs during the algicidal process. The results of this study confirmed that during the algicidal processes of the S3 bacterial culture and sterile supernatant on *A. pacificum*, algal cell lysis led to the continuous release of intracellular PSTs into the water, which resulted in a significant increase in the content of extracellular PSTs. The amount of PSTs released is mainly related to the degree of algal cell lysis. Although approximately 45% of the algal cells were lysed 3 h after treatment with both the 0.5% S3 bacterial culture and sterile supernatant, only small amounts of intracellular PSTs were released because the degree of algal cell lysis was minor at this point (Figure 3C). However, with the progression of the algicidal process, approximately 70% of the algal cells were completely lysed at 12 h (Figure 3F), resulting in an approximately twofold increase in extracellular PSTs over 3 h. Therefore, during the algicidal processes of the S3 bacterial culture and sterile supernatant on *A. pacificum*, PSTs were released slowly in the early stage, and as the algal cells were lysed, the PSTs release process accelerated until the algal cells were completely lysed.

#### 3.3.2. Total PSTs Content, Toxicity, and Profile

The mechanism and regulatory factors of PSTs synthesis in organisms are extremely complex and closely related to environmental factors [55,56,57], genetic differences [52], and the bacteria in the phycosphere [58]. In this study, we found that the contents of intracellular PSTs, extracellular PSTs, and total PSTs did not change significantly in the control and bacterial resuspension treatment groups, while the total PSTs content increased significantly after treatment with the S3 bacterial culture, sterile supernatant, and LB liquid medium. The increase in total PSTs content in the LB liquid medium may be related to the changes in the bacterial community caused by treatment, and the proliferation of *Alteromonas* sp. may have promoted PSTs synthesis [59,60]. In contrast, the increase in the total PSTs contents after treatment with the S3 bacterial culture and sterile supernatant may be caused by the stimulation of residual algal cells by algicidal substances, as the decrease in intracellular PSTs was smaller than the increase in extracellular PSTs during the algicidal process, suggesting that the residual algal cells continue to synthesize toxins during this process. Studies have also demonstrated that *Alexandrium* accelerates toxin synthesis for a survival advantage after exposure to physical, chemical, and biological factors such as low temperature [61], phosphorus limitation [62], high CO_2_ concentration [63], predators [64], and competitors [65]. Therefore, the stress response of residual algal cells to continuous toxin synthesis following stimulation by algicidal substances may be a self-defense mechanism of the algae themselves.

Although small amounts of PSTs were synthesized by *A. pacificum* after algicidal stress with the S3 bacterial culture and sterile supernatant, the total toxicity of the algal sample significantly decreased because of the transformation of the PSTs derivatives. The degradation or transformation of these derivatives can occur through enzymatic reactions (with carbamoylase and sulfotransferase, for example) [10,66,67], bacterial effects [68], or chemical processes [69]. When algal cells were treated with the 0.5% S3 bacterial culture and sterile supernatant, the highly toxic derivatives GTX1/4 were transformed into the less toxic derivatives GTX2/3 and GTX5, resulting in a continuous and significant reduction in the total toxicity of the algal sample (approximately 15% at 12 h). The pathway of reductive transformation of the PSTs GTX1/4 to GTX2/3 after bacterial treatment has also been shown in several studies [70,71,72,73]. In addition, it was speculated that S3 bacterial culture and sterile supernatant could promote the reductive transformation of GTX1/4 to GTX2/3 and may be related to the secretions of S3 since there was no obvious transformation after the algae were treated with the bacterial resuspension and LB liquid medium. Although the types of S3 secretions and their transformation mechanism remain to be investigated, these secretions could be a potential manner to reduce the toxicity of PSTs.

## 4. Materials and Methods

### 4.1. Cultivation of Algae and Bacteria

*A. pacificum* (ATHK), isolated from Jinan University, was cultured in an L1 medium [74] at 20 ± 1 °C with 65 μmol photons/m·s light on a photoperiod of 12 h:12 h (light:dark). The algal cultures were shaken two times per day until the algae reached the late exponential growth phase (a density of 10,000 cells/mL) for the algicidal experiments.

*B. pumilus*, *B. laterosporus*, *B. cereus*, *B. amyloliquefaciens*, *B. subtilis* S2, and *B. subtilis* S3 were purchased from Qingdao GBW Group Co., Ltd. (Qingdao, China), and the strains were cultured in LB liquid medium containing 5 g/L of yeast extract, 10 g/L of peptone, and 10 g/L of NaCl. The incubation temperature was 30 °C, and the rotation speed was 200 r/min. 

### 4.2. Screening of Algicidal Bacteria

*B. pumilus*, *B. laterosporus*, *B. cereus*, *B. amyloliquefaciens*, *B. subtilis* S2, and *B. subtilis* S3 were inoculated into LB liquid medium for 24 h to obtain six kinds of bacterial culture. Then, 0.5 mL of each bacterial culture was added to 50 mL of algal cultures at a final concentration of 1% (*v*/*v*) for 12 h, with untreated algal cultures being used as a control. Then, we fixed each sample with Lugol’s iodine and observed the algae cells under an optical microscope (OLYMPUS CX33, Tokyo, Japan). The numbers of vegetative cells, pellicle cysts, and lysed cells were counted to screen the *Bacillus* sp. with the highest algicidal activity for subsequent experiments, with three replicates for each treatment.

### 4.3. Algicidal Effects of B. subtilis S3

The *B. subtilis* S3 bacterial culture was centrifuged at 8000 r/min for 10 min, and the supernatant was filtered through a 0.22-μm membrane to obtain a sterile supernatant. The bacterial cells were washed three times with sterile seawater and resuspended in an equal volume of sterile seawater to obtain bacterial resuspension. We divided the S3 bacterial culture and its components into four treatment groups, namely, group A (*B. subtilis* S3 bacterial culture), group B (bacterial resuspension), group C (sterile supernatant), and group D (LB liquid medium), with blank algal cultures used as the CK (control). For each treatment group, 0.3%, 0.5%, 1%, and 2% (*v*/*v*) of the corresponding treatment was added to the algal cultures. Then, samples were collected at 0 h, 1 h, 2 h, 3 h, 6 h, 9 h, and 12 h, fixed with Lugol’s iodine, and observed under an optical microscope to count and calculate the algicidal rate, with three replicates for each group.
Algicidal rate (%) = (Nck − Nt)/Nck × 100%
where Nt is the number of intact algal cells in the treatment group at time t, and Nck is the number of intact algal cells in the control group at time t.

### 4.4. Observation of Algal Cell Morphology

Algal cultures were treated with 0.5% (*v*/*v*) S3 bacterial culture, and samples were taken at 0 h, 3 h, 6 h, 9 h, and 12 h to observe the morphological changes in the algal cells under an optical microscope. Another 10 mL of each sample was centrifuged at 8000 r/min for 10 min to collect the algae cells for fixation with 2.5% glutaraldehyde solution for 12 h. Then, the algae cells were washed three times with phosphate-buffered saline (pH 7.2–7.4) and dehydrated with 30%, 50%, 70%, 85%, 95%, and 100% ethanol for 15 min and isoamyl acetate for 4 h. Finally, the algae cell morphology was observed by scanning electron microscopy (Hitachi-S-3400N, Tokyo, Japan) after CO_2_ critical point drying.

### 4.5. Determination of Bacterial Community Structure

#### 4.5.1. DNA Extraction for Sequencing

Group A (*B. subtilis* S3 bacterial culture), group B (bacterial resuspension), group C (sterile supernatant), and group D (LB liquid medium) were treated with 0.5% (*v*/*v*) of the corresponding treatment in algal cultures, and untreated algal cultures were used as a control, with three replicates for each treatment. The four groups were sampled immediately following the addition of the treatment (at 0 h; this sampling time ranged from 2 to 5 min). Then, 10 mL of sample was collected at 0 h, 3 h, and 12 h to determine the bacterial abundance with a flow cytometer (BD FACSCanto, BD Biosciences, San Jose, CA, USA), and an additional 100 mL of sample was filtered through a 0.22-μm membrane. The filter membrane was immediately stored in liquid nitrogen (−80 °C) and analyzed by Novogene Co., Ltd. (Beijing, China). Genomic DNA was extracted from 100 mL of each sample using cetyltrimethylammonium bromide (CTAB) extraction liquid (NobleRyder, Beijing, China). The concentration of extracted DNA was examined using a NanoDrop 2000 spectrophotometer (Thermo Scientific, Waltham, MA, USA). The completeness of DNA samples was checked by electrophoresis in 2% (*w*/*v*) agarose gel. Amplicon sequencing was carried out with the Illumina NovaSeq6000 (Illumina, San Diego, CA, USA). Primer set 341F/806R was used to amplify the 16S rDNA V3-V4 region [75]. Raw sequence data were deposited in NCBI Sequence Read Archive (BioProject accession number PRJNA883237).

#### 4.5.2. Processing of 16S rDNA Genes Sequence Data

Raw paired sequences were uploaded as FASTQ files and analyzed using the Qiime2 pipeline [76]. On the basis of valid data, sequences were merged and denoised with DADA2 and then grouped into amplicon sequence variants (ASVs) at a unique level [77]. To determine the taxonomic lineage of the representative sequence of each ASV, the Classify-learn algorithm of QIIME2 [78,79] was used to annotate the species for each ASV with a pretrained Naive Bayes classifier. According to the ASV annotation results and the characteristics table of each sample, the genus level species abundance table was obtained. Alpha and beta diversity indices were calculated at the ASV level in Qiime2.

### 4.6. Detection of PSTs

#### 4.6.1. Extraction of PSTs 

Group A (*B. subtilis* S3 bacterial culture), group B (bacterial resuspension), group C (sterile supernatant), and group D (LB liquid medium) were treated with 0.5% (*v*/*v*) of the corresponding treatment in algal cultures, and untreated algal cultures were used as a control, with three replicates for each treatment. At 0 h, 3 h, and 12 h, 100 mL of each sample was filtered through a GF/D (Whatman^TM^ glass microfibre filter, diameter 47 mm) membrane. The filter membranes and filtrates were collected for the extraction of both intracellular and extracellular PSTs. The filter membranes were added to 2 mL of 0.1 mol/L acetic acid solution and lysed with a FastPrep-24TM 5G (MP Biomedicals, Santa Ana, CA, USA), and then centrifuged at 5000 r/min for 8 min to collect the supernatant. Then, the supernatant was filtered through a 0.22-μm polyethersulfone membrane (13 mm in diameter) to remove impurities, stored in injection vials, and frozen at −20 °C, protected from light for LC-MS/MS analysis. The filtrates of the sample filtered through the GF/D membrane were enriched with Supelclean^TM^ ENVI-Carb^TM^ 500 mg/6 mL SPE Tube (Reorder NO.57094, Sigma–Aldrich, St. Louis, MO, USA). SPE tubes were activated with 3 mL of acetonitrile and 1 mL of ultrapure water before adding the samples. The samples were naturally drained through the SPE tube and then positively squeezed dry using a Supelco visiprep 24^TM^ DL anti cross contamination SPE device (NO.57265, Sigma–Aldrich, St. Louis, MO, USA). Finally, the samples were eluted and mixed with 1 mL of 75% acetonitrile aqueous solution (containing 0.25% formic acid) and then filtered through a 0.22-μm polyethersulfone membrane into injection vials and frozen at −20 °C protected from light for LC-MS/MS analysis.

#### 4.6.2. LC-MS/MS Analysis

The LC-MS/MS analysis was performed on a 1290 Infinity II UHPLC combined with a G6470A triple quadrupole MS system (Agilent, Santa Clara, CA, USA) equipped with an Agilent jet stream electrospray ionization (AJS-ESI) source. Chromatographic separation of the PSTs was performed using a HILIC-Z column (Poroshell 120 HILIC-Z column; 2.1 mm × 50 mm, 1.9 μm; Agilent). The mobile phase consisted of water (A) and 90% aqueous acetonitrile (B), both containing 2 mM ammonium formate and 0.03% FA (*v*/*v*). The column was maintained at 30 °C, and the injection volume was 10 μL. The gradient elution was as follows, T: 0-1.5-2-9-10-11 min, B%: 100%-100%-100%-30%-20%-100%, flow rate: 0.4-0.4-0.2-0.2-0.2-0.4 mL/min. The equilibration time before the next injection was 5 min. The resolution of the primary mass spectrometry (MS) and secondary mass spectrometry (MS/MS) is in Wide and Unit mode, respectively. The optimized ionization parameters of the AJS-ESI source are as follows: drying gas temperature, 300 °C; sheath gas temperature, 320 °C; drying gas flow rate, 8 L/min; sheath gas flow rate, 11 L/min; nebulizer pressure, 20 psi; capillary voltage, 3500 V in positive and negative acquisition modes. Data were acquired in multiple reaction monitoring (MRM) mode. Data were processed using Agilent Masshunter B 08 software. Other parameters refer to Pan [80]. The HPLC chromatograms of the PSTs standards and one of the samples are shown in Appendix A.

### 4.7. Data Processing

The total content of PSTs (nmol/L) was the sum of the content of each derivative in intracellular and extracellular PSTs.

According to the toxicity equivalent factors (TEF) shown in Table 2 [81], the toxicity of each derivative of PSTs was represented as saxitoxin toxicity equivalent (STX-equ), and the total toxicity of the algal sample was the sum of the toxicity of each derivative.

All data are presented as the mean ± standard error of the mean (SE) in the text and were subjected to a one-way analysis of variance (ANOVA) by Duncan’s post hoc multiple comparison test using SPSS 25.0 for Windows (SPSS, Chicago, IL, USA), with a significance level of *p* < 0.05.

## 5. Conclusions

In conclusion, *B. subtilis* S3 exhibited an efficient algicidal effect on *A. pacificum* by secreting algicidal substances. During the algicidal process, the bacterial community structure in the phycosphere was influenced, which might accelerate algal cell death. This algicidal effect also caused damage to the algal cell structure, resulting in the release of large amounts of intracellular PSTs into the water. Although S3 stimulated *A. pacificum* to synthesize small amounts of PSTs, it promoted the transformation of GTX1/4 to GTX2/3, which reduced the total toxicity of the algal sample. As a consequence, *B. subtilis* S3 may serve as a potential algicidal bacterium that can control HABs caused by the toxic dinoflagellate *A. pacificum*.

## Figures and Tables

**Figure 1 marinedrugs-20-00624-f001:**
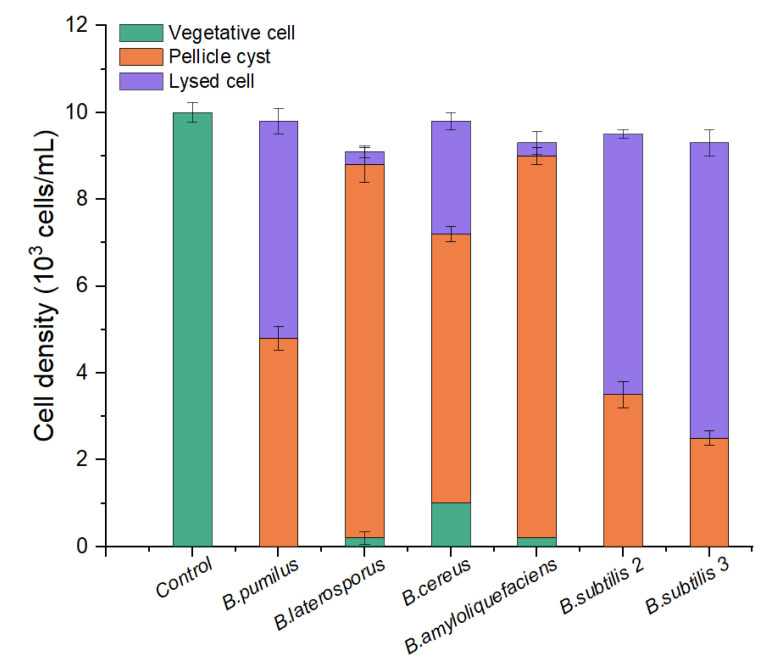
Comparison of the algicidal activity of six strains of *Bacillus* sp. against *A. pacificum*. The control algae culture was treated with 1% (*v*/*v*) LB liquid medium for 12 h, and the treatment groups were treated with 1% (*v*/*v*) bacterial culture of *B. pumilus*, *B. laterosporus*, *B. cereus*, *B. amyloliquefaciens*, *B. subtilis* S2, and *B. subtilis* S3 for 12 h, respectively. Error bars denote standard deviation (*n* = 3).

**Figure 2 marinedrugs-20-00624-f002:**
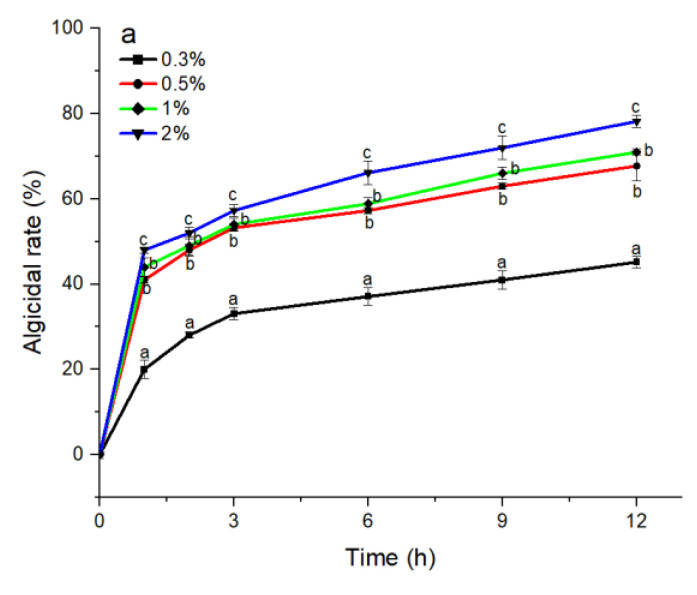
The algicidal activity of 0.3%–2% (*v*/*v*) concentrations of *B. subtilis* S3 bacterial culture (**a**), bacterial suspension (**b**), sterile supernatant (**c**), and LB liquid medium (**d**) against *A. pacificum* at 1 h, 2 h, 3 h, 6 h, 9 h, and 12 h. Error bars denote standard deviation (*n* = 3). One-way ANOVA and Duncan‘s test are used to analyze the difference between treatment groups, and different letters indicate significant differences between different concentrations (*p* < 0.05).

**Figure 3 marinedrugs-20-00624-f003:**
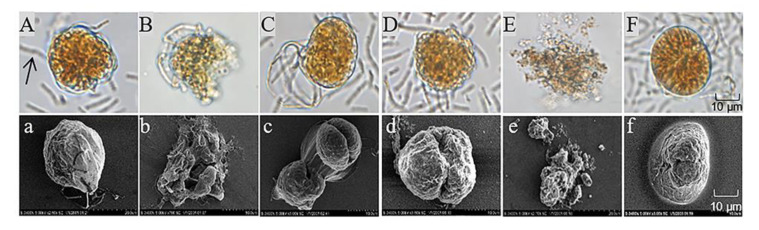
Morphological changes in *A. pacificum* after treatment with *B. subtilis* S3 (S3) bacterial culture (0.5% *v*/*v*). (**A**) Control cells immediately after exposure to S3 at 0 h; (**B**,**C**) cells treated with S3 at 3 h; (**D**,**F**) cells treated with S3 at 6 h; (**E**) cells treated with S3 at 12 h. (**a**–**f**) Corresponding scanning electron micrographs. The black arrow in (**A**) marks S3 cells.

**Figure 4 marinedrugs-20-00624-f004:**
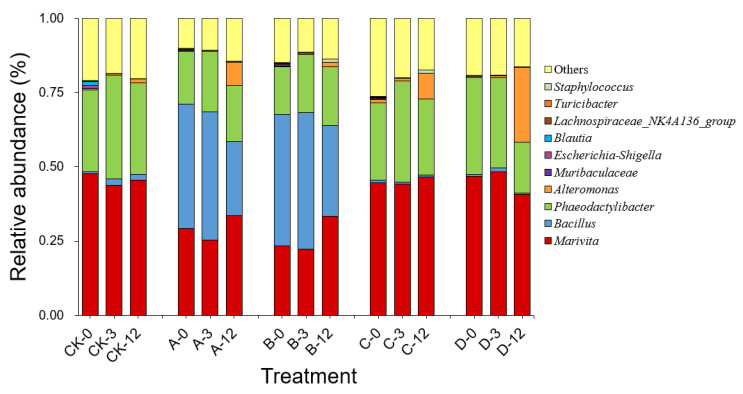
Composition and abundance (genus level) of the top ten bacteria in the bacterial community of *A. pacificum* at 0 h, 3 h, and 12 h of treatment. CK—control algae untreated; A—algae treated with 0.5% (*v*/*v*) *B. subtilis* S3 bacterial culture; B—algae treated with 0.5% (*v*/*v*) bacterial resuspension; C—algae treated with 0.5% (*v*/*v*) sterile supernatant; D—algae treated with 0.5% (*v*/*v*) LB liquid medium.

**Figure 5 marinedrugs-20-00624-f005:**
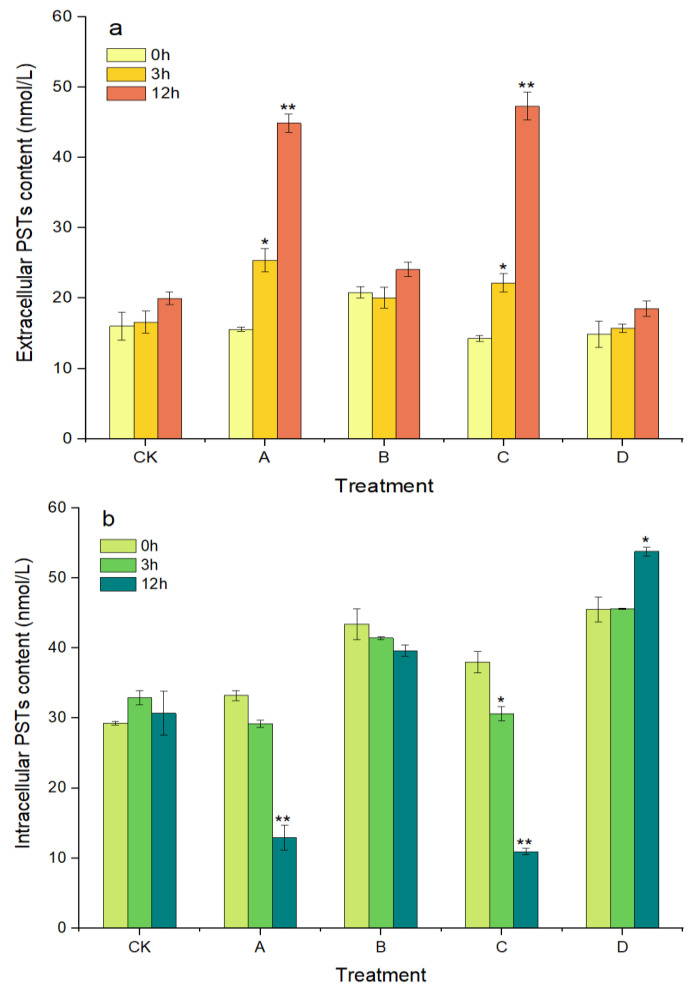
Changes in the content of extracellular PSTs (**a**) and intracellular PSTs (**b**) of *A. pacificum*. CK—control algae untreated; A—algae treated with 0.5% (*v*/*v*) *B. subtilis* S3 bacterial culture; B—algae treated with 0.5% (*v*/*v*) bacterial resuspension; C—algae treated with 0.5% (*v*/*v*) sterile supernatant; D—algae treated with 0.5% (*v*/*v*) LB liquid medium. Error bars denote standard deviation (*n* = 3). One-way ANOVA is used to analyze the difference between each treatment group, and at 0 h, “*” Indicates a significant difference (*p* < 0.05), and “**” indicates an extremely significant difference (*p* < 0.01).

**Figure 6 marinedrugs-20-00624-f006:**
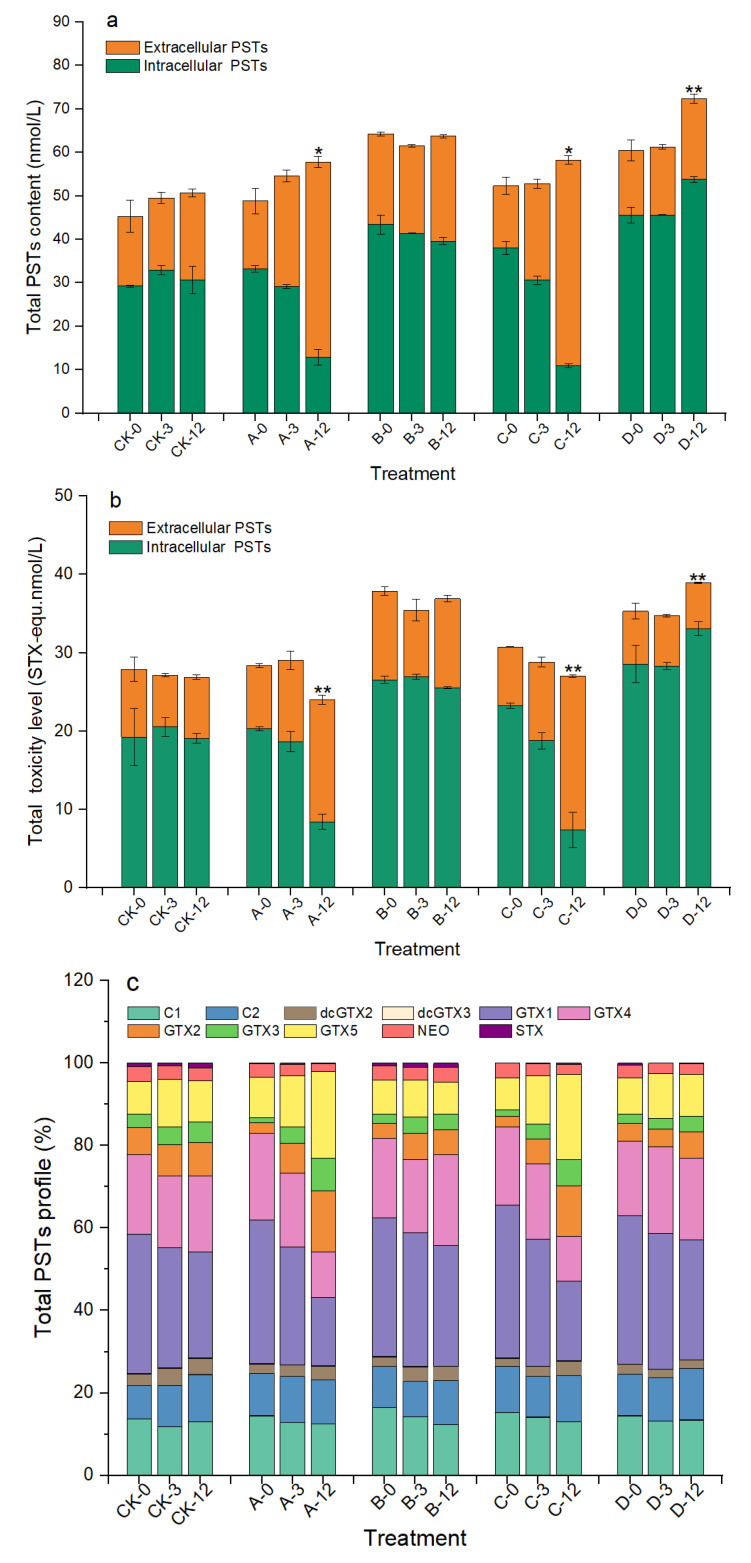
Changes in total PSTs content (**a**), toxicity (**b**), and profile (**c**) of *A. pacificum* at 0 h, 3 h, and 12 h of treatment. CK—control algae untreated; A—algae treated with 0.5% (*v*/*v*) *B. subtilis* S3 bacterial culture; B—algae treated with 0.5% (*v*/*v*) bacterial resuspension; C—algae treated with 0.5% (*v*/*v*) sterile supernatant; D—algae treated with 0.5% (*v*/*v*) LB liquid medium. Error bars denote standard deviation (*n* = 3). One-way ANOVA is used to analyze the difference between each treatment group, and at 0 h; “*” indicates a significant difference (*p* < 0.05), and “**” indicates an extremely significant difference (*p* < 0.01).

**Table 1 marinedrugs-20-00624-t001:** Bacterial abundance and community diversity of *A. pacificum* after 12 h of treatment. (CK) control algae untreated; (A) algae treated with 0.5% (*v*/*v*) *B. subtilis* S3 bacterial culture; (B) algae treated with 0.5% (*v*/*v*) S3 bacterial resuspension; (C) algae treated with 0.5% (*v*/*v*) S3 sterile supernatant; (D) algae treated with 0.5% (*v*/*v*) LB liquid medium.

	Bacterial Abundance (10^7^ Cells/mL)	Shannon	Simpson
CK-12	1.01 ± 0.02	2.80 ± 0.38	0.73 ± 0.03
A-12	2.06 ± 0.11 **	3.43 ± 0.41 **	0.83 ± 0.03 **
B-12	1.46 ± 0.22 *	3.31 ± 0.39 **	0.82 ± 0.03 **
C-12	1.30 ± 0.07	2.94 ± 0.34	0.75 ± 0.03
D-12	1.79 ± 0.19 **	2.91 ± 0.29	0.77 ± 0.02 *

Note: Data represent the mean + SD of triplicate measurements (*n* = 3). One-way ANOVA is used to analyze the difference between the control group and treatment groups, “*” indicates a significant difference (*p* < 0.05), and “**” indicates an extremely significant difference (*p* < 0.01).

**Table 2 marinedrugs-20-00624-t002:** The toxicity equivalent factors (TEF) of PSTs.

PSTs	STX	C1	C2	GTX1	GTX4	GTX2	GTX3	GTX5	dcGTX2	dcGTX3	NEO
TEF	1	0.006	0.096	0.99	0.73	0.36	0.64	0.06	0.833	0.724	0.801

## Data Availability

The datasets generated and analyzed during the current study are available from the corresponding author upon reasonable request.

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
