# Peer review of "A New Perspective: Revealing the Algicidal Properties of Bacillus subtilis to Alexandrium pacificum from Bacterial Communities and Toxins"

_marinedrugs, 2022, doi:10.3390/md20100624_

Round 1
Reviewer 1 Report
The manuscript approaches the algicidal effect of species from the genera Bacillus in Alexandrium pacificum. It goes further, approaching also the effects of exposure to the bacteria Bacillus subtilis S3 in PSTs toxins content and toxicity. This is an important topic, approaching a possible mechanism to control/minimise the effects of HABs and marine biotoxins in the ecosystems and production systems.
Overall the manuscript is well written and easy to understand. However, some major aspects are of concern:
The manuscript should be revised, and definitions and acronyms must be provided the first time they appear, even if described in more detail in the Materials and Methods section.
During the results section, results and discussion are mixed and the Figures and Tables captions are almost all fairly incomplete and unable to stand on their one.
Also due to the side of the manuscript and the amount of data presented, the colouring in figures presenting correlated data should be similar.
More important metodology should be thoroughly revised and described as important information as the number of replicates and samples are missing. Also, this section should provide more detailed information regarding the methodologies since reproducibility may be in question with the current description.
The discussion and conclusion seem to be well-written and oriented. However, an example of a possible practical application would be good.

Author Response
Thank you for your comments and suggestion concerning our manuscript. Those are valuable and very helpful for revising and improving our paper as well as the important guiding significance to our research. We have studied comments carefully and made revisions, which we hope meet with approval.

Reviewer 2 Report
This manuscript reports the results of a laboratory experiment to assess the algicidal potential of several Bacilus strains on toxic dinoflagellates Alexandrium pacificum. The authors also investigated the impact of the algicidal effect on the content and release of paralytic shellfish toxins by the dinoflagellates.
In my opinion, this is an important study worth of publication but substantial improvements are needed to the manuscript:
1- It is important to revise the meaning of some terms and concepts. In the abstract, line 24, it is written “the total toxicity of PSTs decreased”. The authors have not been evaluating the toxicity of the PSP toxins. What was done was determining the concentration of PSP toxins. The sum of each concentration after applying a toxicity equivalence factor may gives the toxicity of the analysed sample but not the toxicity of the toxins. Please revise throughout the manuscript.
2- One of the most important findings of this study is the impact of the algicidal effect on the content and release of PSTs. In section 2.5.2 authors show the changes in PSTs composition and report that some treatments may favor the presence of GTX2/3 and GTX5, and the decrease of GTX1/4. However, it is not clear that such differences are statistically significant, are they?
3- Because in this journal Material and Methods appear at the end of the article, it would be better for the reader to know the meaning of the letters of each treatment. This should be stated in the text and in the legend of the several figures.
4- The impact of the algicidal effect of Bacilus was tested in cultures of Alexandrium, but the authors do not mention the stage of the culture. It is widely known that dinoflagellates may produce and release the toxins differently according to the culture stage. Please state.
5- The section describing the methodology for toxins detection and determination must be substantially improved:
- Page 12, line 393, “The extracellular PSTs were enriched by solid-phase extraction,..” which SPE cartridge was used and how? Only for extracellular PSTs?
- “The processed samples were analyzed by high-performance liquid chromatography (HPLC)”. Please state which equipment and model was used. Which detector? Add the LODs and LOQs
- Page 12, line 396: State the TEFs used for each compound.
- Please provide chromatograms of the toxins analysis in the main text or as supplementary material
Author Response

(The authors gave the same response as above.)

Round 2
Reviewer 1 Report
Paper: "A new perspective: revealing the algicidal properties of Bacillus subtilis to Alexandrium pacificum from bacterial communities and toxins."
The text is, in fact, much improved with the comprehensive revision performed by the authors. There are only two minor points that I would like to see addressed.
1) Please correct the reference on line 47, which currently appears as "[程1]" and cannot be linked to the bibliography;
2) and also add a reference to subsection "4.6.1 Extraction of PSTs" even if it is a repetition of the reference from subsection "4.6.2".
Thank you for your work.
Best regards.
Author Response
Dear reviewer,
We are very grateful to the reviewer for taking the time to conduct the second review and for their valuable suggestions, which significantly improved the quality of the manuscript and allowed us to improve it. According to the reviewer's comment, we have corrected the references on line 47 and section "4.6.1".
Thank you again for the reviewer's help with the manuscript.
Reviewer 2 Report
All issues were well adressed, I can recommend now this manuscript for publication in Marine Drugs
Author Response
Dear reviewer,
We are very grateful to the reviewer for taking the time to conduct the second review and to recommend the manuscript for publication in Marine Drugs.
Thank you again for the reviewer's help with the manuscript.